# The Athletics Injury Prevention Programme Can Help to Reduce the Occurrence at Short Term of Participation Restriction Injury Complaints in Athletics: A Prospective Cohort Study

**DOI:** 10.3390/sports8060084

**Published:** 2020-06-04

**Authors:** Pascal Edouard, Emmanuelle Cugy, Romain Dolin, Nicolas Morel, Jean-Michel Serra, Frédéric Depiesse, Pedro Branco, Kathrin Steffen

**Affiliations:** 1Inter-University Laboratory of Human Movement Science (LIBM EA 7424), University Jean Monnet, University of Lyon, 42023 Saint Etienne, France; 2Department of Clinical and Exercise Physiology, Sports Medicine Unit, Faculty of Medicine, University Hospital of Saint-Etienne, 42055 Saint-Etienne CEDEX 2, France; 3Medical Commission, French Athletics Federation (FFA), 75640 Paris CEDEX 13, France; jean-michel.serra@athle.fr (J.-M.S.); Frederic.DEPIESSE@athle.fr (F.D.); 4European Athletics Medical & Anti Doping Commission, European Athletics Association (EAA), 1007 Lausanne, Switzerland; pedro.branco@european-athletics.org; 5Department of Physical Medicine and Rehabilitation, Hospital of Arcachon, 33260 La Teste de Buch, France; emmanuelle.cugy@ch-arcachon.fr; 6Department of Physical Medicine and Rehabilitation, University Hospital of Bordeaux, 33076 Bordeaux CEDEX, France; 7Handicap Activité Cognition Santé, University of Bordeaux, 33076 Bordeaux CEDEX, France; 8Sportrehab Physiotherapy practice, 34000 Montpellier, France; kinedusportmontpellier@hotmail.fr; 9Department of Orthopeadics, University Hospital of Reims, 51092 Reims CEDEX, France; nicomorel@hotmail.com; 10Physical Medicine and Rehabilitation Department, Sports Medicine Unit, University Hospital of Martinique, 97232 Lamentin, France; 11Oslo Sports Trauma Research Center, Department of Sports Medicine, Norwegian School of Sport Sciences, 0806 Oslo, Norway; kathrin.Steffen@nih.no

**Keywords:** sports injury prevention, injury prevention program, athletics, track and field, epidemiology, prospective studies

## Abstract

We aimed to determine whether an Athletics Injury Prevention Programme (AIPP), targeting the most common athletics injuries, can reduce the occurrence of injury complaints that lead to restrictions in athletics participation (participation restriction injury complaints) in the short (12 weeks) and long (40 weeks) terms. For our 40-week prospective cohort study (level of evidence 2), we invited inter-regional and national-level athletes to regularly perform the AIPP, which included 8 exercises addressing core stability, hamstring, leg and pelvic muscles strengthening and stretching, and balance exercises. A Cox regression was used to analyse the influence of AIPP on the occurrence of participation restriction injury complaint, adjusted to sex, age, height, body mass, discipline, and history of injury complaints during the preceding season, individual response rate, mean weekly training time, mean weekly number of competition, presented by hazard ratio (HR) with 95% confidence interval (95% CI). At 12 weeks (n = 62 athletes), the AIPP was significantly associated with a lower risk of participation restriction injury complaint HR = 0.36 (95% CI: 0.15 to 0.86), *p* = 0.02 and HR = 0.29 (95% CI: 0.12 to 0.73), *p* = 0.009, with cumulative weeks and cumulative training time as time scale, respectively, while at 40 weeks (n = 53 athletes) there was no significant association. An 8-exercise injury prevention programme can effectively help to reduce occurrence of injury complaints that would restrict an athlete’s participation in athletics in the short term.

## 1. Introduction

Athletics (Track and Field) participation leads to a risk of injuries [1]. Indeed, about 61%–76% of athletes incur at least one injury during any one season, and injury incidences from 3.6 to 3.9 injuries per 1000 h of athletics have been reported, varying with age, sex, and athletics disciplines [2,3,4]. Some injuries have been reported to be the most common, with injury diagnosis varying according to the discipline: hamstring muscle injuries (especially in sprints, hurdles, and jumps), Achilles tendinopathies (especially in sprints, jumps, middle- and long-distances), knee overuse injuries including patellar tendinopathies (especially in sprints, middle- and long-distances), shin splints and stress fractures (especially in sprints, middle- and long-distances), ankle sprains (especially in jumps and throws), and low back pain (especially in jumps and throws) [2,3,4,5].

Whatever the level, age or discipline, injury unfortunately affects an athlete’s life. Injuries may force athletes to reduce their training load (volume and/or intensity, so restricting their participation), cause them to drop out of a single competition, miss an entire season or, in the worst case scenario, bring their career to a premature end [1]. Directly or indirectly, injuries negatively influence athletes’ training, performance, and health [1,6,7]. Consequently, injury prevention strategies in athletics represents a win-win perspective for athletes, coaches, and medical staff [1,8,9].

To our knowledge, no injury prevention programme targeting athletics injuries is currently available, contrasting with the situation in team sports [10] (especially football (soccer) [11,12,13] and handball [14,15,16]) which already have scientifically validated injury prevention programmes. These programmes include strengthening exercises and balance and coordination tasks, and have been shown to be effective in reducing injury occurrence, as long as they are completed regularly [10,11,12,13,14,15,16]. Given the specific constraints and injuries associated with athletics practice, developing and analysing the efficacy and/or effectiveness of an injury prevention programme specific to athletics is of interest. In order to cover the greatest possible population at risk of injury caused by athletics practice, such an injury prevention programme should target athletes of competitive level, and not exclusively elite athletes, ranging from youth (under 18 years) to adult (under 40 years) age categories, regardless of sex and discipline.

In this context, the aim of this prospective cohort study was to determine whether an Athletics Injury Prevention Programme (AIPP), implemented in female and male competitive-level athletes ranging from 15 to 40 years, targeting the most common athletics injuries, can reduce the occurrence of injury complaints that lead to restrictions in athletics participation in the short (12 weeks) and long (40 weeks) terms.

## 2. Methods

### 2.1. Study Design

For this 40-week prospective cohort study carried out during the 2014–2015 athletics season, we invited inter-regional and national-level athletes to regularly perform the AIPP. Then, to measure the efficacy of the programme, we asked the athletes to provide us, using a weekly online questionnaire, with data regarding their participation in the programme (i.e., AIPP), any injury complaints incurred and how these affected their participation in athletics, training, and/or competition exposure. We also collected baseline information on each athlete at the start of the season (sex, age, height, body mass, athletic discipline, and history of injury complaints during the preceding season). The study protocol was reviewed and approved by the Saint-Etienne University Hospital Ethics Committee (IORG0004981).

### 2.2. Participants

At the start of the 2014–2015 season, the coaches of 12 athletics training groups from various clubs throughout France were approached. The objectives, procedure, and risks of the study were explained orally (PE) and followed up in writing. A total of 6 of the 12 coaches agreed to participate in this study. In these six training groups, athletes were then selected according to the following inclusion criteria: (1) training at least 3 times a week and engaged in competitive athletics, (2) without any contraindications for competitive athletics activity, and (3) aged from 15 to 40 years. Athletes were included irrespective of their baseline injury status or history [16], because excluding athletes with injury would have resulted in a biased study population not representative of athletes, where injuries are frequent [2,3,4,17]. Informed written consent was obtained from all participants in the study, and in addition from their parents for those under the age of 18 years.

### 2.3. Athletics Injury Prevention Programme and Implementation

The AIPP was developed by a sports medicine physician specialised in athletics (P.E.) with the aim of targeting the most common athletics injuries (hamstring muscle injuries, Achilles and patellar tendinopathies, low back pain, ankle sprains) while being time efficient and feasible. The programme was based on the literature on the epidemiology of athletics injuries, injury risk factors, and current evidence-based injury prevention programmes. Validated exercises used successfully with other cohorts for primary and/or secondary prevention were selected: eccentric strengthening to prevent hamstring injuries [18,19], Achilles tendinopathies [20] and patellar tendinopathies [21]; strengthening and neuromuscular control to prevent ankle sprains [22]; and core stability to guard against low back pain [23]. Before starting the study, the AIPP was reviewed, tested and approved by athletes (n = 7), athletics coaches (n = 4), physiotherapists specialised in athletics (n = 3, including R.D.), and physicians specialised in athletics (E.C., N.M., J.-M.S., F.D.).

The AIPP included 8 exercises with levels of progression (from 2 to 5 depending on the exercise): core stability (plank and side plank), postural control (one-leg balance), pelvic strengthening (lunges and hip abductor strengthening), hamstring exercises (stretching and isometric, concentric and eccentric strengthening), and lower leg exercises (stretching and eccentric strengthening). The exercises, number of repetitions, and levels of progression are presented in Table 1.

The AIPP was sent to all participating athletes and their coaches via e-mail in both paper and video versions, but no further guidance was given. Once the athletes were familiar with the exercises, the AIPP took about 15 min to complete. Athletes were to continue their habitual training and physical conditioning routines and it was left up to them whether and when to do the whole or part of the AIPP programme.

### 2.4. Data Collection Regarding Participation in the AIPP, Exposure and Injury Complaints

Every Monday during the 40-week period of the 2014–2015 season, all the athletes received an invitation by e-mail asking them to complete their online questionnaire to collect information on the preceding week: number of hours of training, participation in competitions (yes/no), number of complete AIPP sessions performed, and injury complaints. Additional e-mails were sent one and three months after the start of the season, encouraging them to fully participate in the study and contribute to the data collection.

An injury complaint was defined as: “A pain, physical complaint or musculoskeletal lesion sustained by an athlete during participation in athletics training or competition, regardless of whether it received medical attention or its consequences with respect to impairments in connection with competition or training” [24]. Injury complaints unrelated to athletics practice were not collected in this study. We chose using the term “injury complaint” since this was self-reported information without medical diagnosis [25].

We asked athletes to report on any injury complaints during the preceding week, using four possible categories [26,27]: no injury complaint, injury complaint allowing full participation, injury complaint necessitating reduced participation, injury complaint preventing any participation.

The primary outcome of the present study was the occurrence of injury complaints that led to restrictions in athletics participation (participation restriction injury complaints), including both injury complaints necessitating reduced participation and injury complaints preventing any participation.

Weekly, we asked athletes to report if they performed the AIPP during the preceding week. We asked athletes to report that they performed an AIPP session only if they had performed the entire programme of 8 exercises. Participation in the AIPP was expressed by the average number of AIPP sessions performed per week.

Finally, an athlete’s participation in the study was expressed by the individual response rate, calculated by dividing the total number of completed weekly questionnaires by the maximum number of responses possible (maximum being 12 or 40 for the short- or long-term analyses, respectively). We only included in the analysis athletes with a response rate ≥75%. No imputations of data were performed.

### 2.5. Data Analysis

Analyses were conducted at 12 and 40 weeks. Descriptive analyses were performed using mean and standard deviation (SD) for continuous variables, and numbers and percentages for ordinal or categorical variables. Normal distribution of the data was verified by the Shapiro–Wilk normality test. Comparisons between groups of athletes included in the analyses at 12 and 40 weeks were performed using *t*-tests for continuous variables and Chi^2^-test for ordinal or categorical variables.

We used a time-to-event approach for analyses [28]. Information after the first occurrence of the primary outcome (i.e., participation restriction injury complaint) was not used in the present analyses (i.e., subsequent participation restriction injury complaints, response rate, AIPP, training and competition); we used the mean number of responses per week, the mean number of AIPP sessions per week, the mean number of training hours per week, and the mean number of competitions per week until the first occurrence of the outcome. Time to first event was analysed using (1) cumulative weeks and then (2) cumulative training time as time scale. The unit of analysis was the individual athlete. A Cox proportional hazards regression (or Cox regression) model was used to analyse the influence of the AIPP on the occurrence of participation restriction injury complaint, adjusted to sex, age, height, body mass, discipline, and history of injury complaints during the preceding season (yes/no), individual response rate, mean weekly training time (except when time scale was cumulative weeks), and mean weekly number of competition. The hazard ratio (HR) with 95% confidence interval (95% CI) was presented for each variable, with the assumption that the HR was constant over time tested.

Significance was accepted at *p* < 0.05. All data were processed using Excel and R (version 3.3.2, ©Copyright 2016 The Foundation for statistical Computing (Comprehensive R Archive Network, http://www.R-project.org)).

## 3. Results

### 3.1. Participants

Of the 140 athletes in the six training groups, 103 (73.6%) athletes met the inclusion criteria and responded at least once to the weekly questionnaire. Among these, 62 (44.3%) athletes (34 male and 28 female athletes) with an average age of 21.4 (SD 5.2) years, achieved a weekly response rate ≥75% at the time of the primary outcome occurrence over the 12-week period, and became the final study group (Table 2). At 40 weeks, 53 (37.9% of the total) athletes had a response rate ≥75% at the time of the primary outcome occurrence over the 40-week period (Table 2). Hence, data from these 62 and 53 athletes were analysed at 12 and 40 weeks, respectively. A flow chart for cohort selection is presented in Figure 1. Analyses of the non-responders did not show any differences in baseline characteristics with either the 103 athletes who met the inclusion criteria or the final study groups.

### 3.2. Short-Term Effect of the AIPP

At 12 weeks, the 62 athletes’ average weekly response rate was 94.3% (SD 8.5, range 75.0–100.0%). A total of 27 (43.5%) athletes presented at least one participation restriction injury complaint over the 12-week period, within a mean time of 5 (SD 4) weeks or 44 (SD 37) hours of athletics training. On average, they completed 1.0 AIPP sessions a week (SD 0.7, range, 0.0–3.4), with no reported injury complaints occurring while performing the AIPP.

The adjusted Cox regression showed that performing the AIPP was significantly associated with lower risk of participation restriction injury complaint (HR = 0.36 (95% CI: 0.15 to 0.86), *p* = 0.02 and HR = 0.29 (95% CI: 0.12 to 0.73), *p* = 0.009, with cumulative weeks and cumulative training time as time scale, respectively) (Table 3). Height, body mass, practicing hurdles, and competitions per week were also significantly associated with risk of participation restriction injury complaint (Table 3).

### 3.3. Long-Term Effect of the AIPP

At 40 weeks, the 53 athletes’ average weekly response rate was 95.7% (SD 7.5, range 75.0%–100.0%). A total of 44 (71.0%) athletes presented at least one participation restriction injury complaint over the 40-week period, within a mean time of 9 (SD 8) weeks or 69 (SD 60) hours of athletics training. On average, they completed 0.9 AIPP sessions a week (SD 0.7, range, 0.0–2.9), with no reported injury complaints occurring while performing the AIPP.

The adjusted Cox regression analysis showed that performing the AIPP was not associated with lower risk of participation restriction injury complaint (*p* > 0.05) (Table 3). Only competitions per week was significantly associated with risk of participation restriction injury complaint (Table 3).

## 4. Discussion

The main finding of the present study was that performing the Athletics Injury Prevention Programme can significantly help to reduce the occurrence of injury complaints leading to restrictions in athletics participation in the short term (during the first 3 months of practice), but not in the long term (over the whole season). These results reveal a protective effect of the AIPP and should be a powerful incentive for athletes and coaches to perform the AIPP regularly.

### 4.1. AIPP Can Reduce Occurrence of Participation Restriction Injury Complaints

Our results show that performing the AIPP can help to reduce the occurrence of participation restriction injury complaints in the short term. These results showed that performing the AIPP can have a rapid preventative effect, since significant reduction was only in the short term. However, it is important to note that other factors were reported to be associated with the risk of participation restriction injury complaints; i.e., height, body mass, practicing hurdles, and competitions per week. This is in agreement with the fact that the nature of injury is multifactorial, and that an injury prevention programme can help to reduce but will probably not prevent injury occurrence.

Our results showed no efficacy of the AIPP to reduce the occurrence of participation restriction injury complaints in the long term. We hypothesise that the present programme was maybe insufficient to improve neuromuscular capabilities of athletes proportionally to the increase in load (i.e., intensity and volume) induced by the athletics season.

### 4.2. Methodological Considerations

A major strength of this study is that it is, to our knowledge, the first to propose and analyse the effect of an injury prevention programme designed specifically for athletics, and is therefore a major step forward with practical implications for athletics injury prevention and athletes’ health [1]. The AIPP is cheap to implement, easy to include in training schedules, easy to perform, and unlikely to harm the athlete.

Another strength is the method used for data collection. This is usually a challenge in athletics, given the training structures and modalities and the organisation of specific medical care for athletes [29]. The method was inspired by those used in athletics by Jacobsson et al. [4] to collect epidemiological injury data through a weekly online questionnaire, as well as those developed by Clarsen et al. [26] to prospectively monitor injuries in elite athletes. We used the injury definition and classifications proposed in the consensus statement of injury and illness definitions and data collection procedures for use in epidemiological studies in athletics [24].

The main limitation of the present study is its design as a prospective cohort study (level of evidence 2), rather than a randomized controlled trial, which is considered the gold standard for evaluating the efficacy of an intervention. However, given the problem of low compliance in randomized controlled trials, which can limit/bias the results, it has been suggested that an alternative approach would be designing prospective cohort studies allowing people to exercise in the way they believe is most appropriate [30], as has been done in the present study.

Participation in a prevention programme could be due to different parameters, such as motivational factors or attitude to injury prevention that can affect both participation in the prevention programme and the outcome (i.e., injury complaint). Such parameters have not been measured in the present study, which can lead to bias in the results.

The sample size of this cohort and the generalisability of the findings also need to be questioned. However, the fact that we have a robust data set for each week based on a high response rate and without data imputations, can explain the small sample size.

As for previous injury prevention programme studies using multiple exercises [12,15], it is not possible to determine precisely which of the exercises were most responsible for the preventive effect, or which exercise is effective for which symptoms or injury complaints. It could have been the sum of all eight exercises that led to the observed preventive effect. Given the multifactorial nature of injuries [31], it should be reasonable to think that an injury prevention programme including different exercises would provide more benefits than an isolated-structure (e.g., muscle or joint) programme [32].

Although we used four categories based on previous studies [26,27] to describe any injury complaints during the preceding week, it could be considered as too simple in comparison to the complex nature of sports injuries [31].

Finally, our lack of quality control when athletes performed the AIPP and on the self-reported data collection [16] needs to be mentioned.

### 4.3. Practical Implications for AIPP and Athlete Health

Compared to other effective injury prevention programs such as that of the FIFA 11+, combining running exercises with strength and conditioning exercises [12,33], the AIPP restricts its action to strength, plyometrics, and balance exercises. As the aim of athletics training is to develop various running skills, running exercises are already a natural part of athletics practice. So, the AIPP focuses on exercises that lead to an improvement in the physical condition of an athlete, and includes exercises that often seem neglected in current athletic training.

Another of our goals was to educate athletes and coaches about the importance of paying attention to their health during athletic training, and to incorporate exercises that protect their health in addition to those focussing on performance. It would be interesting if future projects were to measure whether the AIPP also improves physical performance as we would expect it to be easier to motivate athletes and coaches to follow such an injury prevention programme if there is also a direct performance benefit [34].

From a methodological and sociological perspective, we also think that adding in data collection and analysis parameters related to injury free would be an original and innovative way to show results regarding injuries. Results related to injury prevention are often presented negatively: number of days of absence from sport [24,35,36,37] or the negative consequence on sport participation [37,38]. Highlighting the positive aspects of an injury prevention programme (number of days in good health, with pain-free practice, or with full sport participation…) can help athletes, coaches, and other stakeholders to adhere to injury prevention.

In the present study, we wanted to analyse the effectiveness of the AIPP in the “real life condition”, in which athletes find exercises/programmes on the Internet/YouTube, so no further education was given to coaches or athletes in addition to the paper and video documents. Consequently, our results represent the ecological condition. If we now diffuse this AIPP, we can expect similar benefits for all users.

Finally, the present programme should be improved to be efficient not only in the short term but during the whole season; this could be done by increasing the level of difficulty of the different exercises, developing exercises more appropriate to each specific discipline, individualising the programme to athlete’s specific deficiencies, or including other aspects in addition to neuromuscular improvement such as lifestyle improvement (e.g., nutrition, sleep…).

## 5. Conclusions

This 8-exercise injury prevention programme can help to reduce injury complaints that would restrict their participation in athletics in short term. Given these encouraging results, we suggest using the AIPP in athletics clubs, as a part of athletics training, whatever the sporting level of the athletes. We encourage athletes and coaches to use the AIPP at least once a week especially at the start of the season as a means of preventing injuries and maintaining the integrity of athletes’ health.

## Figures and Tables

**Figure 1 sports-08-00084-f001:**
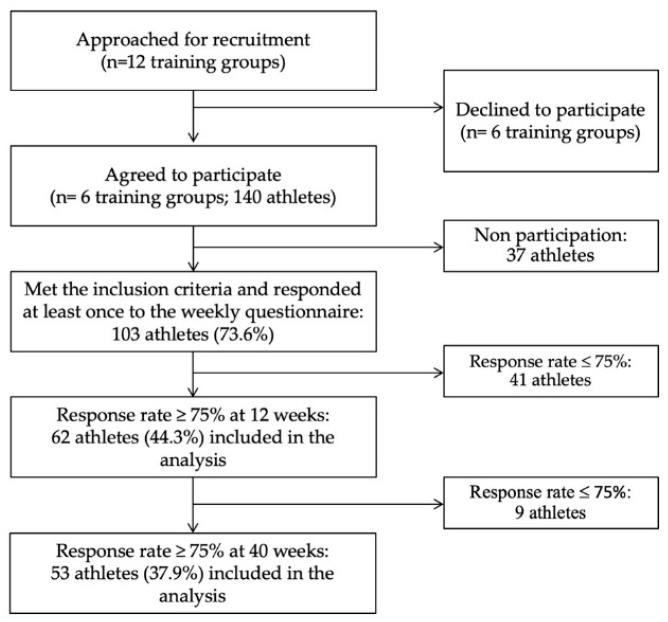
Flow chart showing the recruitment, dropout rate, and number of athletes included and analysed.

**Table 1 sports-08-00084-t001:** The Athletics Injury Prevention Programme used to reduce injury complaints in athletics athletes.

Exercises	Repetitions	Bonus (If It is too Easy, Athlete can Increase Difficulty)
**Core stability—the plank (4 sides: prone position, lateral, supine position, lateral):**		
Level 1: both legs	15 s per side for 3 min	4 × 30 s for 6 min, then for 12 min
Level 2: alternate legs		
Level 3: unstable support		
**Single leg balance:**		
Level 1: static	3 × 15 s (each side)	3 × 30 s each side
Level 2: unstable support		
Level 3: throwing ball with partner, then on unstable support		
**Pelvis strengthening:**		
Lunges	3 × 10 rep.	6 × 10 rep., then 6 × 10 rep. with medicine ball
Hip abductor strengthening-Level 1: leg empty	3 × 10 rep. (each side)	6 × 10 rep. (each side), then unstable support
Hip abductor strengthening-Level 2: with elastic	3 × 10 rep. (each side)	6 × 10 rep. (each side), then unstable support
**Hamstring exercises:**		
Hamstring stretching (different positions)	3 × 15 s	
Hamstring strengthening:		
Level 1: isometric contraction on two legs	6 × 6 s	10 × 10 s
Level 1: heel to buttock with elastic	3 × 6 s (each side)	5 × 10 s (each side)
Level 2: isometric contraction on one leg	6 × 6 s (each side)	10 × 10 s (each side)
Level 2: heel to buttock with elastic	3 × 6 s (each side)	5 × 10 s (each side)
Level 3: Nordic hamstring with help of upper arms	1 × 5 rep., then 3 × 5 rep., then 6 × 6 rep.	
Level 4: Nordic hamstring	1 × 5 rep., then 3 × 5 rep., then 6 × 6 rep.	
Level 5: Pliometric	1 × 5 rep., then 3 × 5 rep., then 6 × 6 rep.	
**Lower leg exercises:**		
Lower leg stretching	3 × 15 s (each side)	
Lower leg strengthening:		
Level 1: down to the ground	3 × 8 rep. (each side)	3 × 10 rep., then 5 × 10 rep. each side and then increase load
Level 2: down to the void	3 × 8 rep. (each side)	3 × 10 rep., then 5 × 10 rep. each side, and then increase loads

s, seconds; rep., repetitions.

**Table 2 sports-08-00084-t002:** Baseline characteristics of the athletics athletes included in the analyses at 12 and 40 weeks. Comparisons between groups at 12 and 40 weeks showed no significant differences.

	At 12 Weeks (n = 62)		At 40 Weeks (n = 53)		Comparison between 12 and 40 Weeks Groups
**Anthropometric data** **(mean (SD))**					
Sex (M/F) (n)	34/28		31/22		Chi2 = 0.16; *p* = 0.69
Age (years)	21.4	(5.3)	23.6	(7.1)	*p* = 0.97
Height (cm)	175.4	(9.3)	175.5	(7.4)	*p* = 0.60
Body mass (kg)	64.3	(10.8)	64.7	(10.0)	*p* = 0.73
Body mass index (kg/m^2^)	20.8	(2.0)	20.9	(2.0)	*p* = 0.89
**History of injury complaints (n (%))**					
In the preceding season	47	(76.8)	41	(77.4)	Chi2 = 0.04; *p* = 0.84
**Disciplines (n (%))**					Chi2 = 1.13; *p* = 0.98
Sprints	17	(27.4)	14	(26.4)	
Hurdles	6	(9.7)	3	(5.7)	
Middle distances	7	(11.3)	5	(9.4)	
Long distances	3	(4.8)	3	(5.7)	
Jumps	6	(9.7)	6	(11.3)	
Throws	3	(4.8)	2	(3.8)	
Combined events	20	(32.3)	20	(37.7)	

M, male athletes; F, female athletes.

**Table 3 sports-08-00084-t003:** Hazard ratio (HR) with 95% confidence interval (95% CI) of the association of the Athletics Injury Prevention Programme (AIPP) and other factors with the occurrence of participation restriction injury complaints, according to the time scale (cumulative weeks and cumulative training time), at 12 and 40 weeks.

	Time Scale: Cumulative Weeks				Time Scale: Cumulative Training Time			
	HR	95% CI	*p* Value	*Time-Independancies of Variables*	HR	95% CI	*p* Value	*Time-Independancies of Variables*
**At 12 weeks**								
**Main factor: AIPP**	**0.36**	**(0.15 to 0.86)**	**0.02**	*0.95*	**0.29**	**(0.12 to 0.73)**	**0.009**	*0.80*
Adjustment variables:								
Sex (M)	1.35	(0.17 to 10.7)	0.78	*0.94*	1.34	(0.16 to 11.5)	0.79	*0.62*
Age	1.01	(0.99 to 1.21)	0.09	*0.97*	1.10	(1.00 to 1.21)	0.06	*0.09*
Height	**1.91**	**(1.03 to 3.53)**	**0.04**	*0.98*	**2.08**	**(1.09 to 3.99)**	**0.03**	*0.76*
Body mass	**0.43**	**(0.20 to 0.98)**	**0.04**	*0.96*	**0.40**	**(0.17 to 0.92)**	**0.03**	*0.83*
Body mass index	8.11	(0.67 to 98.6)	0.10	*0.91*	11.2	(0.78 to 159.0)	0.08	*0.60*
Disciplines (reference combined events):								
Sprints	0.51	(0.13 to 2.01)	0.33	*0.053*	0.64	(0.16 to 2.56)	0.53	*0.33*
Hurdles	**0.01**	**(0.01 to 0.95)**	**0.05**	*0.29*	**0.01**	**(0.01 to 0.96)**	**0.05**	*0.82*
Middle distances	0.53	(0.09 to 3.14)	0.48	*0.80*	0.54	(0.08 to 3.41)	0.51	*0.24*
Long distances	0.39	(0.02 to 9.68)	0.57	*0.44*	0.62	(0.02 to 15.2)	0.77	*0.63*
Jumps	1.29	(0.23 to 7.32)	0.78	*0.26*	1.66	(0.31 to 8.90)	0.56	*0.33*
Throws	0.00	(0.00 to inf)	1.00	*1.00*	0.00	(0.00 to inf)	1.00	*1.00*
History of injury complaints in the preceding season	1.40	(0.33 to 6.03)	0.65	*0.86*	1.53	(0.35 to 6.67)	0.57	*0.30*
Response rate	1.05	(0.98 to 1.13)	0.17	*0.58*	1.03	(0.95 to 1.10)	0.57	*0.50*
Training per week	1.14	(0.93 to 1.39)	0.20	*0.25*				
Competition per week	**37.7**	**(4.79 to 295.9)**	**0.0006**	*0.15*	**48.3**	**(6.76 to 344.3)**	**0.0001**	*0.91*
*Cox model adjustment*	*Concordance* = *0.824* (*se* = *0.038*); *Likelihood ratio test* = *39.55 on 16 df*, *p* < *0.0001*; *Wald test* = *29.92 on 16 df*, *p* = *0.002*; *Score* (*logrank*) *test* = *38.69 on 16 df*, *p* = *0.001*				*Concordance* = *0.838* (*se* = *0.034*); *Likelihood ratio test* = *43.63 on 15 df*, *p* = *0.0001*; *Wald test* = *32.92 on 15 df*, *p* = *0.005*; *Score* (*logrank*) *test* = *47.61 on 15 df*, *p* < *0.0001*			
*Time-independancies of variables*				*chi-2 test* = *14.9*; *p* = *0.53*				*chi-2 test* = *15.7*; *p* = *0.40*
**At 40 weeks**								
**Main factor: AIPP**	0.79	(0.47 to 1.31)	0.36	*0.88*	0.87	(0.52 to 1.44)	0.58	*0.99*
Adjustment variables:								
Sex (M)	0.98	(0.22 to 4.35)	0.98	*0.93*	1.00	(0.24 to 4.21)	1.00	*0.86*
Age	0.97	(0.89 to 1.06)	0.50	*0.43*	1.04	(0.95 to 1.15)	0.38	*0.67*
Height	1.63	(0.90 to 2.94)	0.11	*0.65*	1.19	(0.65 to 2.17)	0.57	*0.46*
Body mass	0.50	(0.23 to 1.09)	0.08	*0.63*	0.76	(0.35 to 1.65)	0.49	*0.44*
Body mass index	7.27	(0.68 to 77.7)	0.10	*0.64*	2.13	(0.19 to 23.7)	0.54	*0.44*
Disciplines:								
Combined events	1.70	(0.10 to 32.4)	0.72	*0.49*	0.32	(0.02 to 4.21)	0.41	*0.71*
Sprints	2.31	(0.12 to 44.2)	0.58	*0.30*	0.35	(0.03 to 4.94)	0.44	*0.53*
Hurdles	8.68	(0.31 to 247.1)	0.21	*0.57*	0.89	(0.05 to 16.9)	0.94	*0.87*
Middle distances	1.95	(0.10 to 36.2)	0.66	*0.51*	0.57	(0.04 to 8.83)	0.69	*0.90*
Long distances	4.75	(0.16 to 144.8)	0.37	*0.42*	0.94	(0.03 to 29.6)	0.97	*0.23*
Jumps	10.3	(0.50 to 215.3	0.13	*0.90*	2.90	(0.18 to 47.4)	0.46	*0.86*
Throws	0.00	(0.00 to inf)	1.00	*1.00*	0.00	(0.00 to inf)	1.00	*1.00*
History of injury complaints in the preceding season	0.81	(0.26 to 2.48)	0.71	*0.83*	1.36	(0.45 to 4.13)	0.59	*0.92*
Response rate	**1.08**	**(1.01 to 1.15)**	**0.03**	*0.32*	1.03	(0.98 to 1.09)	0.26	*0.66*
Training per week	9.10	(0.78 to 1.06)	0.23	*0.03*				
Competition per week	**0.54**	**(0.43 to 0.69)**	**<0.0001**	*0.38*	**0.61**	**(0.48 to 0.76)**	**<0.0001**	*0.63*
*Cox model adjustment*	*Concordance* = *0.876 (se* = *0.026)*; *Likelihood ratio test* = *87.75 on 17 df, p* < *0.0001*; *Wald test* = *41.89 on 17 df*, *p* < *0.0001*; *Score* (*logrank*) *test = 73.95 on 17 df*, *p* < *0.0001*				*Concordance* = *0.84* (*se* = *0.033*); *Likelihood ratio test* = *78.91 on 16 df*, *p* < *0.0001*; *Wald test* = *40.81 on 16 df*, *p* < *0.0001*; *Score* (*logrank*) *test* = *68.46 on 16 df*, *p* < *0.0001*			
*Time-independancies of variables*				*chi-2 test* = *10.6; p* = *0.88*				*chi-2 test* = *85.4*; *p* = *0.93*

AIPP, Athletics Injury Prevention Programme; Inf, Infinity; Significant results are highlighted in bold.

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
