# Peer review of "The Athletics Injury Prevention Programme Can Help to Reduce the Occurrence at Short Term of Participation Restriction Injury Complaints in Athletics: A Prospective Cohort Study"

_sports, 2020, doi:10.3390/sports8060084_

Round 1

Reviewer 1 Report

please see the attached file.

Author Response

REVIEWER #1

Ms. Ref. No.: Manuscriptsports-738690

Title:"The athletics injury prevention programme can help to reduce the prevalence of participation restriction injury complaints in athletics: a prospective cohort study."

Sports

 Revised Manuscript: Revision #1                                                                                                  

Reviewers' comments:

This prospective cohort study aimed to understand the long- and short- term effect of an intervention of the Athletics Injury Prevention Programme (AIPP) on injury prevention. This research takes a lot of time to complete.

We would like to thank the reviewer for taking his/her time to review our manuscript, providing this supportive comment, and seeing some clinical implications forAthletics injury prevention.

However, the prevention program did not focus one symptom or injury. It compose core stability, balance, pelvic strengthening, hamstring and leg exercise. That would result in that the program can't distinguish which exercise is effective for which symptoms or injury. Only tell us it can reduced the injury or complaints. Those may put in the “limitation” to explanation.

Thank you for this relevant comment. We agree that our prevention programme did not focus one symptom or injury, and our aim was to target the most common athletics injuries: hamstring muscle injuries, Achilles and patellar tendinopathies, low back pain, ankle sprains. Therefore, we included 8 exercises. We agree with this limitation, that has been added, with an explanation, Page 12 lines 141-147: “As for previous injury prevention programme studies using multiple exercises[12,15], it is not possible to determine precisely which of the exercises were most responsible for the preventive effect, or which exercise is effective for which symptoms or injury complaints. It could have been the sum of all eight exercises that led to the observed preventive effect. Given the multifactorial nature of injuries [31], it should be reasonable to think that an injury prevention programme including different exercises would provide more benefits than a isolated-structure (e.g. muscle or joint) programme [32].”.

In addition, the weekly prevalence only divided 4 categories, I thought that category is quick of too simple. This part can be explained more. But, overall, the quality of this manuscript is good.

Thank you for this relevant comment. Indeed, each week, athletes should reply to a query and reported if they had on any injury complaints during the preceding week. To reply athletes can chose among 4 categories: no injury complaint, injury complaint allowing full participation, injury complaint necessitating reduced participation, injury complaint preventing any participation. These categories have been developed based on a previous study on athletics injury data collection (Edouard et al. 2015), and inspired from Clarsen et al. 2014 suggesting a new approach for injury data collection. The “participation restriction injury complaints” included both reduced participation and zero participation injury complaints, and was chosen because such injury complaints negatively affect the athlete’s practice. This is indeed a restrictive and simple view of the injury problem that meet athletes, but as the present study represents a first step in the injury prevention, we think that such classification and our primary outcome represent a relevant aspect in practice.

According to your comment, we added a sentence in the limitation section Page 12 lines 148-150: “Although we used four categoriesbased on previous studies [26,27]to describe any injury complaints during the preceding week, it could be considered as too simple in comparison to the complex nature of sports injuries[31].”.

Reviewer 2 Report

Very nice paper, well presented and I hope interesting to the readers.

Reviewer.

Author Response

REVIEWER #2

Ms. Ref. No.: Manuscriptsports-738690

Title:"The athletics injury prevention programme can help to reduce the prevalence of participation restriction injury complaints in athletics: a prospective cohort study."

Sports

 Revised Manuscript: Revision #1  

Reviewers' comments:

Very nice paper, well presented and I hope interesting to the readers.

We would like to thank the reviewer for taking his/her time to review our manuscript, providing this supportive comment, and seeing some clinical implications forAthletics injury prevention.

Reviewer 3 Report

In their study on the effect of an injury prevention program to prevent injuries that limit sports participation in athletes, the authors present a novel evaluation of a prevention programme. There are, however, some concerns with the study design and statistical approach.

Compliance is not a good way to divide participants into exposed vs. non-exposed groups. Compliance with the prevention programme could be due to motivational factors or attitude to injury prevention. This has not been measured (as far as I can tell) and therefore cannot be controlled for in the analysis. This creates confounding: attitude to injury prevention can affect both participation in the programme and the outcome (injuries).

Athletes may have stopped or reduced participation in the injury prevention programme due to injuries. This would have created bias in the sample (greater number of injuries in those who were ‘less compliant’); it is not clear if this has been taken into account in the analysis. Athletes may also have dropped out of the study because of injury.

This study requires fully adjusted statistical modelling, to adjust for baseline factors and potential confounders. Given the current design and statistical analysis limitations, a causal effect of the prevention programme and subsequent injury occurrence cannot be established.

Minor comments:

In the introduction, please specify the setting of the study (professional vs. amateur; setting; athletics events; age range).

In table 2, please provide statistical significance of the comparisons between groups (anthropometric data, etc.)

Author Response

REVIEWER #3

Ms. Ref. No.: Manuscriptsports-738690

Title:"The athletics injury prevention programme can help to reduce the prevalence of participation restriction injury complaints in athletics: a prospective cohort study."

Sports

 Revised Manuscript: Revision #1  

Reviewers' comments:

In their study on the effect of an injury prevention program to prevent injuries that limit sports participation in athletes, the authors present a novel evaluation of a prevention programme. There are, however, some concerns with the study design and statistical approach.

We would like to thank the reviewer for taking his/her time to review our manuscript and provide these comments, which help to improve our present manuscript.

Compliance is not a good way to divide participants into exposed vs. non-exposed groups. Compliance with the prevention programme could be due to motivational factors or attitude to injury prevention. This has not been measured (as far as I can tell) and therefore cannot be controlled for in the analysis. This creates confounding: attitude to injury prevention can affect both participation in the programme and the outcome (injuries).

Athletes may have stopped or reduced participation in the injury prevention programme due to injuries. This would have created bias in the sample (greater number of injuries in those who were ‘less compliant’); it is not clear if this has been taken into account in the analysis. Athletes may also have dropped out of the study because of injury.

This study requires fully adjusted statistical modelling, to adjust for baseline factors and potential confounders. Given the current design and statistical analysis limitations, a causal effect of the prevention programme and subsequent injury occurrence cannot be established.

Thank you for this very relevant comment. We initially chose to analyse the efficacy of the prevention programme based on the compliance to the programme and using a priori cutoff of 1 per week which can be relevant for practical implications. However, we do agree with your comments regarding the confounding induced by compliance. In addition, a time-to-event approach would be more relevant to avoid the limit induced by subsequent injuries and the consequence of injuries on the compliance with the injury prevention programme. Although in our previous analyses we reported no significant differences between groups, we do agree that adjust the analyses to recorded factors is important.

Therefore, and following your very relevant comment, we reconducted the statistical analysis using a time-to-event approach. The methods and results sections have been consequently rewritten, and the discussion adapted to the new results.

Minor comments:

In the introduction, please specify the setting of the study (professional vs. amateur; setting; athletics events; age range).

Following your comment, we have clarified the setting of the study in the introduction Page 2 lines 107-111: “In order to cover the greatest possible population at risk of injury caused by athletics practice, such injury prevention programme should target athletes of competitive level, and not exclusively elite athletes, ranging from youth (under 18 years) to senior (under 40 years) age categories, regardless of sex and disciplines.”. This additional information can help to better support the inclusion and exclusion criteria presented in the methods section (Page 3 lines 135-144).

In table 2, please provide statistical significance of the comparisons between groups (anthropometric data, etc.)

Following your comments regarding the statistical data analyses, we change the statistical data analyses, the two groups no longer exist; Table 2 has been modified to present the included population in analysis at 12 and 40 weeks, and comparison of these two groups of athletes was made and statistical significance added.

Round 2

Reviewer 1 Report

The researchers answer the questions well. 

Reviewer 3 Report

The authors have made improvements to the manuscript by using proportional hazard modelling to analyse the outcome: injuries. However, I have still have two concerns with the study. 

First, my previous comment has not been addressed: 

Compliance is not a good way to divide participants into exposed vs. non-exposed groups. Compliance with the prevention programme could be due to motivational factors or attitude to injury prevention. This has not been measured (as far as I can tell) and therefore cannot be controlled for in the analysis. This creates confounding: attitude to injury prevention can affect both participation in the programme and the outcome (injuries).

Second, I could not quite understand how the proportional hazards modelling was done. The outcome was occurrence of first injury (since starting the program). However, the variable of interest would need to be participation in the AIPP, at any given timepoint. As AIPP participation changes over time, this should be introduced as a time-dependent variable (although this creates the problems outlined in my first point). Also, the results show a different effect of AIPP participation on injury early vs. late in the program, therefore the hazards are not proportional. Please clarify the modelling approach and revise as necessary. 

Author Response

REVIEWER #3

Ms. Ref. No.: Manuscriptsports-738690_revised

Title: "The athletics injury prevention programme can help to reduce the occurrence at short term of participation restriction injury complaints in athletics: a prospective cohort study."

Sports

 Revised Manuscript: Revision #2  

Reviewers' comments:

The authors have made improvements to the manuscript by using proportional hazard modelling to analyse the outcome: injuries. However, I have still have two concerns with the study. 

We would like to thank the reviewer for his/her supportive comment and help to improve our present manuscript.

First, my previous comment has not been addressed: 

Compliance is not a good way to divide participants into exposed vs. non-exposed groups. Compliance with the prevention programme could be due to motivational factors or attitude to injury prevention. This has not been measured (as far as I can tell) and therefore cannot be controlled for in the analysis. This creates confounding: attitude to injury prevention can affect both participation in the programme and the outcome (injuries).

Please accept our apologies, indeed there were no specific response for this comment, the response was integrated with the next comment. And we have tried to do our best to follow your relevant comment by changing the analysis: we move from a comparison analysis between two groups based on the compliance with the prevention programme, to a time-to-event analysis using the participation to the prevention programme as an explanatory variable.

However, this is indeed a very relevant comment, which deserves a more specific reply. We agree that the compliance with the prevention programme could be due to several factors, such as motivational factors or attitude to injury prevention, and that such factors can create confounding since these can affect both participation in the programme and the outcome (injuries). Indeed, these factors have not been measured in our present study, and we therefore can not controlled for these confounding factors in the present analyses. A way to deal with this limit would be to use an instrumental variable analysis (Greenland 2000). Such analytical approach allows to deal with unmeasured confounding factors. However, in our present study we did not find any parameter, which meet the assumptions of an instrumental variable (Martens et al. 2006, Hernan et al. 2006, Kjaesgaard et al. 2016), and allow us to use an instrumental variable analysis. Thus, following your relevant comment on these confounding factors, we added a sentence in the discussion section to present this limitation (Page 11 lines 48-51): “Participation in a prevention programme could be due to different parameters, such as motivational factors or attitude to injury prevention that can affect both participation in the prevention programme and the outcome (i.e. injury complaint). Such parameters have not been measured in the present study, that can lead to bias the results.”. We hope that this specific response meets your expectations, if no, we appreciate if you can provide some clues to achieve this.

Second, I could not quite understand how the proportional hazards modelling was done. The outcome was occurrence of first injury (since starting the program). However, the variable of interest would need to be participation in the AIPP, at any given timepoint. As AIPP participation changes over time, this should be introduced as a time-dependent variable (although this creates the problems outlined in my first point). Also, the results show a different effect of AIPP participation on injury early vs. late in the program, therefore the hazards are not proportional. Please clarify the modelling approach and revise as necessary. 

Thank you for asking us to clarify the modelling approach. We used a time-to-event approach for analyses (Nielsen et al. 2019). Information after the first occurrence of the primary outcome (i.e. participation restriction injury complaint) was not used in the present analyses (i.e. subsequent participation restriction injury complaints, response rate, AIPP, training and competition). For AIPP, training and competition, we used as values the mean number of AIPP sessions per week, the mean number of training hours per week and the mean number of competitions per week until the first occurrence of the outcome. Thus, AIPP, training and competition performed after the occurrence of the primary outcome was not used in the analysis (as for time-to-event approach). This has been clarified in the manuscript (Page 5 lines 73-78): “We used a time-to-event approach for analyses [28]. Information after the first occurrence of the primary outcome (i.e. participation restriction injury complaint) was not used in the present analyses (i.e. subsequent participation restriction injury complaints, response rate, AIPP, training and competition); we used the mean number of responses per week, the mean number of AIPP sessions per week, the mean number of training hours per week and the mean number of competitions per week until the first occurrence of the outcome.”.

In addition, we analysed the time-independancies of variables. The results of the global analysis were previously reported in the Table 3., reporting no time-independancies of variables. Given your present comment, we added the values for each variable, the Table 3 has thus been updated (Page 8). The time-independancies analysis of the variable AIPP was insignificant (p-values from 0.80 to 0.99), confirming that the variable AIPP was not a time-dependent variable.

Wereported that performing the Athletics Injury Prevention Programme was significantly associated with a reduced occurrence of injury complaints leading restrictions to athletics participation in the short term (during the first 3 months of practice, analysis at 12 weeks), but not in the long term (over the whole season, analysis at 40 weeks). Since time-independancies of variables was insignificant, such different in the results between the short and the long terms can not be explained by a bias lead by the statistical analysis. Therefore, we hypothesised that this difference could be explained by neuromuscular factors. Indeed, we hypothesized that the present programme was maybe insufficient to improve neuromuscular capabilities of athletes proportionally to the increase in load (i.e. intensity and volume) induced by the athletics season.